# Predicting pK$_a$ for proteins using COSMO-RS

Martin Peter Andersson[1], Jan Halborg Jensen[2] and
Susan Louise Svane Stipp[1]

[1] Nano-Science Center, Department of Chemistry, University of Copenhagen, Copenhagen,
DK-2100, Denmark
[2] Department of Chemistry, University of Copenhagen, Copenhagen, DK-2100, Denmark

## ABSTRACT

We have used the COSMO-RS implicit solvation method to calculate the equilibrium constants, pK$_a$, for deprotonation of the acidic residues of the ovomucoid inhibitor protein, OMTKY3. The root mean square error for comparison with experimental data is only 0.5 pH units and the maximum error 0.8 pH units. The results show that the accuracy of pK$_a$ prediction using COSMO-RS is as good for large biomolecules as it is for smaller inorganic and organic acids and that the method compares very well to previous pK$_a$ predictions of the OMTKY3 protein using Quantum Mechanics/Molecular Mechanics. Our approach works well for systems of about 1000 atoms or less, which makes it useful for small proteins as well as for investigating portions of larger proteins such as active sites in enzymes.

## INTRODUCTION

Proteins are the basic building blocks of life as we know it. Better understanding of their chemical and physical behavior would open a host of new possibilities in science, medicine and technology. Central to understanding their behavior is a method for describing their properties thermodynamically. The protonation state is one important variable for predicting interaction with fluids and solids because as pH changes and protons attach or detach from the protein as a function of pH and solution composition, charge and adhesion properties are affected. The equilibrium constant that describes protonation, the acidity constant, pK$_a$, provides a quantification of the protein's properties and contributes to our ability to predict the outcome of processes such as protein–protein interaction (*Muegge, Schweins & Warshel, 1998*; *Sheinerman, Norel & Honig, 2000*), aggregation (*Wang, Li & Speaker, 2010*) and interactions with nanoparticles (*Bomboi et al., 2013*) and surfaces. These processes in turn control biological activity. Considerable effort has gone into research on pK$_a$, to determine values experimentally as well as developing and validating methods to predict them. Many of these are described in a recent review (*Alexov et al., 2011*). An interesting point throughout is that pK$_a$ values for protonation of amino acid side chains can be significantly shifted from their water reference values. For predicting acidity constants, several approaches have been used: quantum chemical,

Corresponding author
Martin Peter Andersson,
ma@nano.ku.dk

**Peer**J

molecular dynamics, electrostatic (Poisson–Boltzmann and generalized Born) and empirical methods. The empirical methods often outperform more rigorous methods. In a recent blind prediction study (*Olsson, 2012*), the largest deviation from experimental data resulted from a quantum chemical method. Much of the difficulty in reducing uncertainty in protein acidity constants comes from difficulties with configurational sampling, which is crucial for capturing details about structural reorganization and water penetration into the protein.

The turkey ovomucoid inhibitor protein, OMTKY3, has frequently been used as a system for benchmarking $pK_a$ predictions in proteins, in particular quantum mechanical (QM) and combined quantum mechanical/molecular mechanical (QM/MM) methods. Accurate measurements of its five acid residues are available (*Schaller & Robertson, 1995*) and there is a significant spread between the most and the least acidic values. This spread, coupled with the reasonable size of the protein, makes OMTKY3 a very good model for benchmarking theoretical results (*Li, Robertson & Jensen, 2004*).

$pK_a$ prediction, using density functional theory (DFT) and the implicit solvent model COSMO-RS, is quite straightforward for acids (*Klamt et al., 2003*) as well as bases (*Eckert & Klamt, 2006*) and does not require any explicit solvent molecules. The prediction of $pK_a$ values for small inorganic and organic molecules using COSMO-RS is mature, with calculated values matching experimental values with a root mean square error of 0.5 pH units, but how well could $pK_a$ values be predicted when an acid group is part of a protein and both the acid group and the rest of the protein are affected by internal hydrogen bonding and local changes in the environment? In this study, we demonstrate that given a reasonable starting structure based on experimental evidence, a combination of semi-empirical geometry optimization, coupled with single point calculation using DFT, and the implicit solvent model COSMO-RS, gives excellent agreement for the $pK_a$ values of the five acid side chains in OMTKY3. We also provide evidence that the accuracy of protein $pK_a$ predictions using COSMO-RS is equally good for larger biomolecules such as whole or parts of protein molecules that consist of as many as 1000 atoms.

## COMPUTATIONAL DETAILS

Geometry optimization was performed using the program MOPAC2009 (*Stewart, 2009*) with the AM1 (*Dewar et al., 1985*) and PM6-DH+ (*Korth, 2011*) semi-empirical methods and the linear scaling algorithm MOZYME. The COSMO solvent module with dielectric constant 999.9 was used for the geometry optimization because the COSMO-RS method requires perfect screening as the reference state. We used the LBFGS method for geometry optimization, with GNORM = 2.0. All DFT simulations were single point calculations performed with the TURBOMOLE program, v6.3[3] (*Ahlrichs et al., 1989*), using the BP functional (*Becke, 1988*; *Perdew, 1986*) and the SVP basis set (*Schafer, Horn & Ahlrichs, 1992*; *Weigend & Ahlrichs, 2005*). The COSMOtherm program with parameterization BP_SVP_C21_0111 was used for all COSMO-RS calculations (*Eckert & Klamt, 2002*; *Eckert & Klamt, 2013*) and all were performed at 298 K. $pK_a$ calculations in COSMO-RS are based on a linear free energy relationship between measured $pK_a$ and the calculated free energy

[3] TURBOMOLE V6.3 2011, a development of University of Karlsruhe and Forschungszentrum Karlsruhe GmbH, 1989–2007, TURBOMOLE GmbH, since 2007.

difference between the protonated and the deprotonated forms of the acid (*Klamt et al., 2003*). There is no need for explicit solvent molecules.

Our starting point for the geometry optimization for the OMTKY3 protein was the experimental structure for 1PPF (*Bode et al., 1986*) downloaded from the PDB databank. The initial positions of the hydrogen atoms were determined at pH 7, using the WHATIF program (*Vriend, 1990*). We used two approaches for geometry optimization: We first used the AM1 method, with full optimization of the whole protein for each of the protonated states. As a second approach, we used the PM6-DH+ method analogously but we had to make an additional geometry optimization. Before the single point DFT calculations, we froze the entire structure except for the amino acid side chain of interest and re-optimized using AM1. This is a prerequisite for using the COSMO-RS parameterization BP_SVP_C21_0111, for which only the AM1 semi-empirical method gives reliable results.

The initial structure was determined for neutral pH, which means that all aspartic acid, Asp, and glutamic acid, Glu, were deprotonated and negatively charged, whereas lysine, Lys, was protonated and positively charged. All $pK_a$ values were then predicted by adding a single hydrogen atom to each acid group in turn, using only the most stable conformation found. At least two conformations were tested for each acid group by adding a hydrogen atom to either of the acid oxygen atoms. No conformer treatment was made in COSMO-RS.

The calculation time was quite reasonable for a small (805 atoms) protein. The semi-empirical geometry optimization of the experimental structure took about 5 h on a single core. The next geometry optimizations, where protons were added, took less than an hour each. A single point DFT calculation took about 15 h (wall time) using 8 cores.

## RESULTS AND DISCUSSION

The optimized geometry and corresponding COSMO surface for the OMTKY3 protein is shown in Fig. 1. The predicted $pK_a$ values for the acid side chains of OMTKY3 match very well with experimental data (Table 1) with an RMS error of less than 1 pH unit for both optimization methods. This compares well with previous studies (*Forsyth et al., 1998*; *Havranek & Harbury, 1999*; *Li, Robertson & Jensen, 2004*; *Mehler & Guarnieri, 1999*; *Nielsen & Vriend, 2001*). The PM6-DH+ optimization, where RMS error is 0.5, gives slightly better results than AM1, where RMS is 0.8. This improvement probably comes from the much more accurate treatment of internal hydrogen bonding with the PM6-DH+ method, which results in a better three-dimensional protein structure. Particularly important is the ability of our method to predict the most acidic $pK_a$ value of Asp27 with an error of less than 1 pH unit. The excellent agreement implies that $pK_a$ prediction using COSMO-RS is as good for macromolecules as it is for smaller inorganic and organic compounds, for which the method is parameterized. COSMO-RS has been shown to be as accurate for small bases (*Eckert & Klamt, 2006*) as it is for small acids (*Klamt et al., 2003*), which suggests that COSMO-RS could also be applied to $pK_a$ predictions for base side chains of proteins.

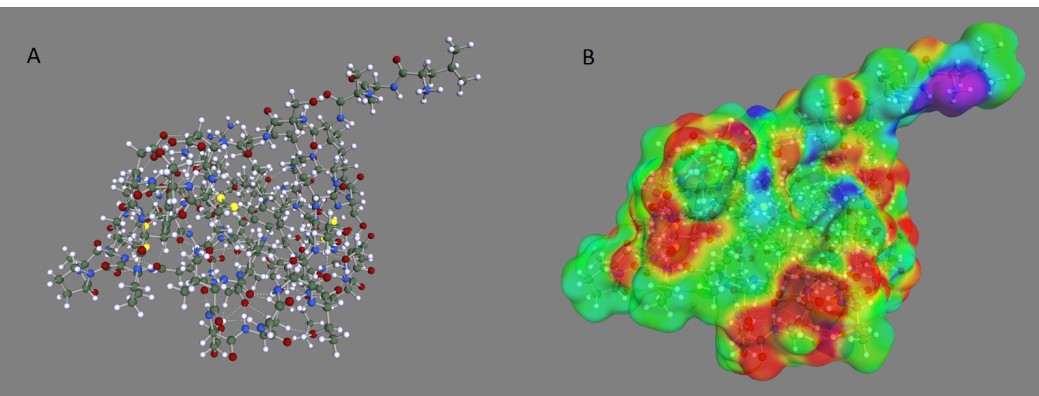

**Figure 1 Optimized geometry and COSMO surface of the OMTKY3 protein.** (A) Optimized geometry and (B) COSMO surface of the OMTKY3 protein. The PM6-DH+ semi-empirical method was used for the geometry optimization and the BP/SVP/COSMO method was used to create the COSMO surface.

**Table 1 Experimental and calculated pK$_a$ values for the acids in OMTKY3.** The method in parenthesis is the method used for the geometry optimization.

| Amino acid side chain | Experimental data (*Schaller & Robertson, 1995*) | This work (AM1) | This work (PM6-DH+ /AM1) | *Li, Robertson & Jensen (2004)* | *Forsyth et al. (1998)* | *Nielsen & Vriend (2001)* | *Mehler & Guarnieri (1999)* | *Havranek & Harbury (1999)* |
|---|---|---|---|---|---|---|---|---|
| Asp7 | 2.5 | 2.7 | 2.2 | 2.4 | 2.9 | 2.7 | 2.9 | 2.1 |
| Glu10 | 4.1 | 5.7 | 3.8 | 4.3 | 3.4 | 3.6 | 4.1 | 4.0 |
| Glu19 | 3.2 | 3.5 | 2.6 | 2.7 | 3.2 | 2.7 | 3.6 | 3.1 |
| Asp27 | 2.2 | 2.6 | 3.0 | 1.9 | 4.0 | 3.4 | 3.3 | 2.9 |
| Glu43 | 4.8 | 4.9 | 4.6 | 4.5 | 4.3 | 4.3 | 4.4 | 5.6 |
| | | | | | | | | |
| RMS error | | 0.8 | 0.5 | 0.3 | 0.9 | 0.7 | 0.6 | 0.5 |
| Max error | | 1.6 | 0.8 | 0.5 | 1.8 | 1.2 | 1.1 | 0.8 |

OMTKY3 is a small protein but our results suggest that parts of larger proteins, with as many as 1000 atoms (about 50 amino acid residues), for example, active sites in enzymes, could be predicted in a similar manner. This would require slightly more complex methods where one might freeze atoms for parts of the protein that are far from the site of interest. Assuming an average protein density of 1.22 g/cm$^3$ (*Andersson & Hovmöller, 1998*) and using polyglycine as an average protein composition and structure, we can construct a model protein with ∼1000 atoms contained within a sphere of radius 13.8 Å. This system is large enough to represent most active sites in enzymes and similar molecules of interest. To use a pure QM method, the molecular structure must be terminated outside the sphere, i.e., by removing atoms more than 13 Å from the center of interest and adding protons to any broken bonds, similar to the approach followed by Li and colleagues (*2004*). By choosing a QM treatment for a system of as many as 1000 atoms, we can significantly reduce the artifacts that would accompany the need for a QM/MM boundary for a smaller number of atoms.

The COSMO-RS method, using semi-empirical geometry optimization, has an accuracy that is close to much more elaborate QM/MM methods but the computational setup and cost are significantly smaller. For proteins of <1000 atoms, application of the method is quite straightforward because the whole structure can be considered, with no need to cut bonds. It would be advantageous to have a parameterization of COSMO-RS that is based on geometry from a more accurate semi-empirical method than AM1. From our results, we suggest that the PM6-DH+ method could provide such a framework, considering its good performance for biomolecules (*Yilmazer & Korth, 2013*).

## CONCLUSIONS

We have predicted $pK_a$ for the OMTKY3 protein that is quite close to experimental data. The COSMO-RS implicit solvent model works very well for proteins, where internal hydrogen bonding and local environment modify the $pK_a$ values from what they would be for the free amino acids. The root mean square error for the five acidic side chains in the OMTKY3 protein was 0.5 pH units, which is comparable to results from previous efforts to predict $pK_a$. Our approach works well for systems of about 1000 atoms or less, which makes it useful for small model proteins and for investigating portions of larger proteins such as active sites in enzymes.

### Funding
The computing resources were provided on a system established by the Danish Center for Scientific Computing (DCSC), which is now known as DeIC (Danish e-infrastructure cooperation). This work is part of the Materials Interface with Biology (MIB) consortium, which is funded through an UK EPSRC grant (EP/I001514/1). The funders had no role in study design, data collection and analysis, decision to publish, or preparation of the manuscript.

### Grant Disclosures
The following grant information was disclosed by the authors:
UK EPSRC: EP/I001514/1.

### Competing Interests
The authors state that they have no competing interests.

### Author Contributions
- Martin Peter Andersson conceived and designed the experiments, performed the experiments, analyzed the data, contributed reagents/materials/analysis tools, wrote the paper.
- Jan Halborg Jensen analyzed the data, wrote the paper.
- Susan Louise Svane Stipp wrote the paper.

## Supplemental Information

Supplemental information for this article can be found online at http://dx.doi.org/10.7717/peerj.198.

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
