# Peer review of "Predicting pKa for proteins using COSMO-RS"

_PeerJ, doi:10.7717/peerj.198_

## Round 0.1 · original submission · Minor Revisions

Please address critical points raised by the reviewer 2.

Reviewer 1 ·

Basic reporting

No comments

Experimental design

No comments

Validity of the findings

Better consider larger set of experimental pKa's. But as "proof of concept" is it ok as it is.

Comments for the author

No comments

Reviewer 2 ·

Basic reporting

Figures should be revised to include legend with detailed description of the content.

Experimental design

No comments

Validity of the findings

No comments

Comments for the author

The manuscript entitled” Predicting pKa for proteins using COSMO-RS” by Andersson et al. described the application of COSMO RS salvation method to predict pKa values of the acidic residues of the protein, OMTKY3, yielding well-matched pKa values with the measured values and the predicted values from other simulation methods. Given the overall qualities of the manuscript, I would like to recommend its publications after the authors address the following minor revisions.

1) The authors claimed that the COSMO-RS method for prediction of pKa worked well with system of ~1000 atoms. This statement could be clearer to readers by converting the atom number to amino acid number, even in molecular weights.

2) This work was focused on predicting pKa values for acidic residues. Therefore the discussion of predicting pKa for basic residues in line 91 should be revised to avoid over-interpretation. Such as by stating that “COSMO-RS could also be applied to pKa predictions for base side chain”

3) Figures should include legends with detailed description that allows readers to follow without referring to text.

4) Figures 1 nicely presents the pka prediction data in color-coded surface of the OMTKY3 protein. I would suggest to present the surface of OMTKY3 protein using measured pka data for side-by-side, visual comparison.

---

## Round 0.2 · accepted · Accept

All critical points were addressed and the manuscript was revised accordingly.